# Comparison Between Nano-Hydroxyapatite/Beta-Tricalcium Phosphate Composite and Autogenous Bone Graft in Bone Regeneration Applications: Biochemical Mechanisms and Morphological Analysis

**DOI:** 10.3390/ijms26010052

**Published:** 2024-12-24

**Authors:** Igor da Silva Brum, Lucio Frigo, Jemima Fuentes Ribeiro da Silva, Bianca Torres Ciambarella, Ana Lucia Rosa Nascimento, Mario José dos Santos Pereira, Carlos Nelson Elias, Jorge José de Carvalho

**Affiliations:** 1Department of Implantology, School of Dentistry, State University of Rio de Janeiro, 157, 28 de Setembro, Boulevard, Rio de Janeiro 20551-030, Brazil; mario.pereira@uerj.br; 2Department of Basic Sciences, Faculdade de Odontologia da Associação Paulista de Cirurgiões Dentistas, 457, Voluntarios da Patria, St. São Paulo 02011-000, Brazil; lucio.frigo@faoa.edu.br; 3Laboratory of Ultrastructure and Tissue Biology, Department of Histology and Embryology, State University of Rio de Janeiro, Rio de Janeiro 20550-900, Brazil; jemima.fuentes@uerj.br (J.F.R.d.S.); ciambarella.bianca@ce.uerj.br (B.T.C.); ana.nascimento@uerj.br (A.L.R.N.); 4Instituto Militar de Engenharia, 80, Praça Gen, Tiburcio, Rio de Janeiro 22290-270, Brazil; elias@ime.eb.br; 5Department of Biology, School of Medicine, State University of Rio de Janeiro, Professor Manuel de Abreu, 444, Avenue, Rio de Janeiro 20550-170, Brazil; carvalho@uerj.br

**Keywords:** bone graft, autogenous bone, biomaterial, hydroxyapatite, beta-tricalcium phosphate

## Abstract

It was assumed that only autogenous bone had appropriate osteoconductive and osteoindutive properties for bone regeneration, but this assumption has been challenged. Many studies have shown that synthetic biomaterials must be considered as the best choice for guided bone regeneration. The objective of this work is to compare the performances of nanohydroxyapatite/β-tricalcium phosphate (n-HA/β-TCP) composite and autogenous bone grafting in bone regeneration applications. The composite was characterized by scanning electron microscopy (SEM) and used as an allograft in bone defects formed in adult Wistar rats. The bone defects in the dorsal cranium were grafted with autogenous bone on one side and the n-HA/β-TCP composite on the other. Histomorphometry evaluation via different staining methods (Goldner trichrome, PAS, and Sirius red) and TRAP histochemistry were performed. Immunohistomorphometries of OPN, Cathepsin K, TRAP, acid phosphatase, VEGF, NFκ-β, MMP-2, MMP-9, and TGF-β were carried out. The RT-PCR method was also applied to to *RANK-L*, *Osteocalcine*, *Alcaline Phosphatase*, *Osterix*, and *Runx2*. The results showed that for all morphometric evaluations with the different staining methods, histochemistry, and immunohistochemistry, VEGF and NFκ-β were higher in the n-HA/β-TCP composite group than in the autogenous bone graft group. The RT-PCR markers were higher in the autogenous bone group than in the n-HA/β-TCP composite group. The n-HA/β-TCP composite exhibited enhanced cell–matrix interactions in bone remodeling, higher adhesion, proliferation, and differentiation, and increased vascularization. These results suggest that the n-HA/β-TCP composite induces faster bone formation than autogenous bone grafting.

## 1. Introduction

The research and development of biomaterials has been pressed to design new products and to improve the ones already available due to the growing demand for bone grafts and substitutes. Bone grafts and substitutes are used in surgical procedures when a bone fracture occurs or when there is bone loss. Bone loss and reduced bone density occur due to factors such as old age and degenerative diseases. The literature estimates that 2.2 million clinical procedures involving bone grafts are registered annually [1]. The market size for global bone grafts and substitutes was valued at USD 2.80 billion in 2022, and an expected annual growth rate of 6.2% up to 2030 is expected [2]. The increased demand is related to severe bone injury and bone loss increase due to trauma, tumor, infections, surgical resections, and some age-related diseases [3].

Some authors stated that the ideal biomaterial for bone grafts should exhibit the following biological properties: (1) biocompatibility, meaning the biomaterial and its metabolites should be nontoxic; (2) biodegradability, meaning it should be completely absorbable after achieving its goals; (3) bioplasticity, meaning it should adopt the desired shape and respond to the mechanical demands of the tissue; (4) osteoinductivity, meaning it should provide differentiation of mesenchymal stem cells to osteoprogenitor cells and promote bone formation at heterotopic places; (5) osteoconductivity, meaning it should provide a 3D scaffold able to support blood vessel sprouting and osteoprogenitor cells anchoring to new bone formations; and (6) sterilizable, meaning it should be suitable for common sterilization procedures without reducing its biological properties [4,5,6].

The search and design of a ideal biomaterial ought to be preceded by a comprehensive understanding of bone formation, remodeling, and healing mechanisms. Detailed and extensive descriptions of bone physiology provide a solid background to biomaterial design and evaluation [7,8,9].

Bone healing is a regenerative process. The mechanism starts with the formation of new bone tissue and, subsequently, its remodeling. The process is based on cell migration, differentiation, extracellular matrix production, and mineralization [10,11].

Clot formation after bone injury and the inflammation process which follows are responsible for the release of a myriad of cell mediators related to cell migration and differentiation [12]. Bone morphogenic protein 4 (BMP4), interleukine-6 (IL-6), vascular endothelial growth factor (VEGF), platelet-derived growth factor (PDGF), and transforming growth factor-β (TGF-β) are important mediators in the process. The mediators induce the recruitment of mesenchymal stem cells (MSCs), commitment to the osteoprogenitor cell line, osteoblast differentiation, osteoclast precursor recruitment, and vascular sprouting [13,14,15,16,17,18,19].

At the intracellular level, Runt-related transcription factor 2 (Runx2), SOX9, and Osterix (OSX) gene expression mark the commitment to mature osteoblasts. Nuclear factor κB (NFκB), c-fos, tumor necrosis factor (TNF) receptor-associated factor 6 (TRAF6), and sequestosome 1 (SQSTM1) are transcription factors required for osteoclast differentiation. The release of the notch intracellular domain (NCID) by the γ-secretase complex and presenilin 1 (PS1) or PS2 is linked to type H endothelial cells in the revascularization site [20,21,22,23,24,25].

The first report of biomaterial development (synthetic calcium phosphate) dates back to the 1930s. In the 1960s and 1970s, biopolymers and bioglasses were designed, and in 1985, the description of the BMPs further propelled biomaterials’ design. In 1990 and 2000, the union of calcium phosphate with peptides and different types of materials was witnessed. However, for more than a decade, hydroxyapatite was not evident, until the appearance of the hydroxyapatite/β-TCP composite. More recently, nanotechnology has enhanced the osteoinductive and osteoconductive properties of biomaterials [26] (Figure 1).

In the case of extensive bone loss. A bone graft is usually used to fill the defect and guide new bone formation (osteoconduction). According to the source, bone grafts can be classified as (1) autografts, where the donor and receptor are the same individual, (2) isografts, where the donor is an identical twin, (3) allografts, where the donor is from the same species, (4) xenografts, where the donor is from a different species, and (5) synthetic, where biomaterials similar to bone are used [27,28]. Autogenous bone is considered the “gold standard” due to is potential to trigger osteoinduction and osteoconduction. However, autogenous bone grafting has limitations related to the extension of bone loss and morbidity in the site from where the bone is extracted [29,30]. Nanotechnology has stimulated a search for a synthetic biomaterial which could rival osteoinduction and osteoconduction of autogenous grafts.

The aim of this work is to compare the osteoinduction performance of n-HA/β-TCP composites and autogenous bone use as grafts in critical bone defects in rats.

## 2. Results

### 2.1. X-Ray Diffraction

The diffraction pattern of the nano-HA/ß-TCP composite showed narrow, high, and well-defined peaks, which suggest excellent crystallinity (Figure 2).

In the Figure 3, the three phases are detailed: β-Ca_3_(PO_4_)_2_ with trigonal symmetry (Figure 3A), Ca_10_(PO_4_)_3_(OH) with hexagonal symmetry (Figure 3B), and Ca_10_(PO_4_)_3_(OH) with monoclinic symmetry (Figure 3C). The Ca_10_(PO_4_)_3_(OH) monoclinic phase had the highest concentration. The Ca_10_(PO_4_)_3_(OH) hexagonal phase appeared only as a trace. These results indicate that processing of the n-HA/ß-TCP compound induced the transformation of the hexagonal phase to become monoclinic.

### 2.2. Goldner’s Trichromic Staining

Goldner’s Masson trichrome staining was applied to the regenerated bone structures. The critical defect treated with Blue Bone (B) was more suffused with newly formed bone than the autogenous group (A), as can be seen by the increase in red areas in the image, indicating the osteoid bone matrix. Mineralized bone formation can be observed in green (Figure 4).

The red-stained osteoid matrix was more evident in the lining of the bone marrow, the blood vessels, and the biomaterial. Group differences were related to the amount of red staining (Figure 5).

### 2.3. Picrosirius Red (PSR)

A PSR and HE method comparison better indicated the bone defect regions (Figure 6).

The PSR slides indicate in pink the synthetized collagenous osteoid bone matrix (Figure 6A,C) and the same microscopy field in the polarized microscopy method (Figure 6B,D). Figure 7 shows the percentage of osteoid bone matrix bone and collagenous areas stained by polarized picrosirius red.

The picrosirius red staining with polarized microscopic observation can be used to distinguish the maturation (thickness and packing) of type 1 collagen in the bone tissue [31]. The yellow and red stains represent mature type 1 collagen, and green represents immature type 1 collage. The bone of calvaria (asterisk) exhibited a green color with areas of mature collagen (yellow). In critical defects (star), a higher immature collagen (green) content may be noted indicating bone regeneration.

### 2.4. Cathepsin K

Cathepsin K is an osteoclastic intracellular activity marker downstream of RANK-L and NFκ-β. Cathepsin K is responsible for the degradation of type I collagen in osteoclast-mediated bone resorption.

In Figure 8, the dark brown immunostained slide indicates cathepsin K’s immunoreaction in the autogenous graft (Figure 8A) and n-HA/β-TCP composite (Figure 8B). Figure 9 shows the percentage of osteoclast regulation through osteoblast activity by cathepsin K.

### 2.5. *Transforming Growth Factor Beta* (TGF-β)

The TGF-β induced osteoclasts apoptosis, blocking bone matrix reabsorption and osteoblast recruitment. The dark brown immunostaining indicates TGF-β’s immunoreactivity in the autogenous bone group (Figure 10A) and n-HA/β-TCP composite (Figure 10B). Figure 11 shows the results of the histomorphometric analysis of TGF-β immunostaining.

TGF-B had significant osteoinductive activity in vivo, stimulating mesenchymal precursor cells to differentiate into osteoblasts and matrix formation. In group B, intense and diffuse staining in the extracellular matrix was observed, being more intense in the mesenchymal cells (asterisks), which may suggest the induction of new bone formation (Figure 11).

### 2.6. Osteopontin (OPN)

Figure 12 shows the osteopontin-stained slides’ photomicrography. The slides showed dark brown deposits mostly found scattered in the decalcified bone matrix. Some round-nucleus cells (probably osteoblasts) presented high-intensity deposits in the cytoplasm. These round-nucleus immunoreactive cells were found in higher numbers near the biomaterial groups. Figure 13 shows the area percentage of OPN in the samples. The stages of bone neoformation in the central region of the critical defects were evaluated through immunostaining for osteopontin (OP). OP is a marker of immature bone tissue, and it was observed in the extracellular matrix in group B, showing the beginning of the bone formation process.

### 2.7. Vascular Endothelial Growth Factor (VEGF)

Vascular endothelial growth factor (VEGF) immunoreactivity was observed mainly in the cells’ cytoplasm close to and in the blood vessels in all tested groups. Only the n-HA/β-TCP composite group (Figure 14B) showed the presence of VEGF-ir. VEGF-ir was found in the round-nucleus cells scattered around the biomaterial particles and inside the degrading biomaterial as well. Figure 15 shows the percentage area of VEGF-ir in each group. In both groups, positive VEGF expression was also observed in the blood vessels, extracellular matrix, and bone cells. However, an increased number of osteocytes were immunostained for VEGF.

### 2.8. Nuclear Factor Kappa-Beta (NFκ-β)

NFκ-β regulates TNF-α and RANK-L, hence inducing osteoclast differentiation (Figure 16). In the immunohistochemistry for nuclear factor kappa beta (NFKB) in Figure 16A, indicated by the arrow, we can see exact immunolabeling in an osteoclast. In Figure 16B, we can see immunolabeled islands, showing regulation of the osteoclasts.

Figure 17 shows the area percentage of NFκ-β-ir in each group. In both groups, positive NFκ-β expression was also observed in the blood vessels, extracellular matrix, and bone cells.

### 2.9. Metalloproteinase-9 (MMP-9)

MMP-9 was observed in dark brown deposits inside the osteoblasts and in the bone matrix (Figure 18). Decreased MMP-9 levels can influence the bending strength and toughness of bone because this affects ECM proteins. In group B, the most intense staining was noted in the extracellular matrix, osteocytes (cross), and fibroblast-like cells, showing more of an influence on bone strength. Figure 19 shows the area percentage of MMP-9 in the samples, suggesting higher expression of MMP-9 in the Blue Bone group.

### 2.10. MMP-2 (Metalloproteinase-2)

The dark brown deposits indicate MMP-2-ir in the osteoblasts and bone matrix (Figure 20).

MMP-2 is a gelatinase related to bone integrity because deficiency-affecting mineral ion exchanges occur during resorption and mineralization of the bone matrix. In the B group, it was noted that the most intense staining in the extracellular matrix, osteocytes, and fibroblast-like cells showed more potential in maintaining bone mineral density [32] (Figure 20). Figure 21 shows the area percentage of MMP-2 in the samples, suggesting a higher enzyme content.

### 2.11. Schiff’s Periodic Acid (PAS)

The PAS method is based on Schiff’s periodic acid oxidizing hydroxyl and amino-alkyl amine chemical groups, forming a magenta-colored complex. It detected polysaccharides, glycoproteins, and glycolipids, suggesting new bone formation (Figure 22).

PAS presents a positive reaction with mineralized bone, while it does not react with osteoid tissue [33]. The results showed thick PAS-positive trabeculae in the Blue Bone group (REG) and a single thin trabecula in the autogenous group (arrows), suggesting the osteogenic potential of the Blue Bone group (REG) (Figure 23).

### 2.12. Bone Morphogenic Protein-2 (BMP-2)

BMP-2 is an extremely important signal in the process of bone remodeling because it regulates the differentiation of mesenchymal cells into osteoblasts, maintaining the stability of the process. The asterisks show the regions in darker brown, which were considered to be intense markings revealing the presence of osteoblasts and the osteoid matrix (Figure 24).

The Mann–Whitney test showed that no significant statistical relevance was found in the Blue Bone group in relation to the autogenous group (Figure 25).

### 2.13. Real-Time Polimerase Chain Reaction (RT-qPCR)

The RT-qPCR data on *RANKL* gene expression indicates no difference between groups. On the other hand, *OSTEOCALCIN*, *ALKALINE PHOSPHATASE*, *OSTERIX*, and *RUNX2* showed significant reductions in gene expression in the n-HA/β-TCP group (Figure 26).

### 2.14. Validation Test for Analysis of Acid Phosphatase (AP) Activity in Bone

The validation of the AP assay kit shows it to be possible to evaluate acid phosphatase activity (Figure 27A) and non-TRAP activity (Figure 27B) in plasma, as well as in the rats’ bone tissue. Although the AP activity showed a significantly decrease in bone (0.0034 ± 0.002608) when compared with plasma (0.0108 ± 0.002775), those results were probably caused by the difference in the amount of protein of each sample. The non-tartrate-resistant acid phosphatase activity (non-TRAP activity) showed no significant difference between plasma and bone, indicating that the amount of AP derived from osteoclasts was similar in both plasma (0.00454 ± 0.003627) and bone (0.00158 ± 0.001117).

### 2.15. Acid Phosphatase Assay

For both AP activity (Figure 28A) and non-TRAP activity (Figure 28B), no difference was observed between the acid phosphatase derived from the bone control group (0.0590 ± 0.03851; 0.0288 ± 0.01968) and the group grafted with autogenous bone (0.0662 ± 0.01465; 0.0332 ± 0.006979). The group grafted with n-HA/β-TCP presented a significant increase in AP activity (0.2074 ± 0.09938) and non-TRAP activity (0.1018 ± 0.04779) compared with the control group and the group grafted with autogenous bone. Although the non-TRAP activity was similar to 50% of the total AP activity, these results suggest increased remodeling of the bone by activation of the osteoclasts.

### 2.16. MEV (Scanning Electron Microscopy)

Figure 29 shows the SEM images. It is possible to observe structures related to the bone remodeling process, like bone matrix trabeculae and biopolymers in addition to osteoblasts and osteoclasts.

## 3. Discussion

Bone tissue’s morphophysiology and response to trauma and disease is a flourishing field of research. It received additional importance to support the design and manufacturing of graft biomaterials [34].

In this study, we provided morpho-physiological evidence that a synthetic biomaterial has the potential to enhance more natural responses to trauma than autogenous bone. In our comparison between a synthetic biomaterial (n-HA/ß-TCP composite) and autogenous grafts, it was found that (1) in the histomorphometric evaluation (Goldner’s trichromic, Schiff’s periodic acid, tartrate-resistant acid phosphatase, and Picrosirius red), the n-HA/ß-TCP composite presented more new bone formation areas than autogenous bone; (2) in the phosphatase alkaline assay, the n-HA/ß-TCP composite showed higher activity than the autogenous bone; and (3) in the immunohistochemistry evaluation (cathepsin K, TGF-β, osteopontin, VEGF, NFκ-β, MMP-2, and MMP-9), the n-HA/ß-TCP composite presented more immunoreactive areas than the autogenous bone. On the other hand, the Rt-PCR analysis (*OSTEOCALCIN*, *ALKALINE PHOSPHATASE*, *OSTERIX*, and *RUNX2*) showed higher gene expression in the autogenous bone group but not higher *RANKL* gene expression (Figure 26).

Goldner’s Masson staining is considered one of the best staining techniques to distinguish between a new bone matrix and mature bone matrix [35]. Picrosirius red staining is more focused on the identification of fibrillar collagen networks [36]. Additionally, Shiff’s periodic acid is the choice staining method for detecting polysaccharides, glycoproteins, and glycolipids in bone matrices [37]. Taken together, the more extensive bone matrix production in the n-HA/ß-TCP composite group suggests faster production of the organic matrix component than in the autogenous bone group. The MEV photomicrographs highlight collagen bundles and fibrils in the n-HA/ß-TCP composite group.

Acid phosphatase (AP) induces inorganic phosphate release from the mineralization inhibitor inorganic pyrophosphate (PPi) in the bone matrix. Tartarate-resistant acid phosphatase (TRAP) exhibits potent phosphatase activity on osteopontin (OPN), and its dephosphorylation reduces OPN’s inhibitory mineralization potential [38,39,40].

Despite finding higher immunoreactivity for OPN, higher activities were found for acid phosphatase (AP) and tartarate-resistant acid phosphatase (TRAP) in the n-HA/ß-TCP composite group, indicating a greater mineralization potential according to the proposal by Linder et al. [39].

Cathepsin K is mainly found in osteoclasts, the bone cell type related to bone matrix absorbance, and its gene expression is regulated by RANKL-RANK transduction signaling (NFκ-β). Cathepsin K is highly efficient at breaking down type I collagen and osteonectin, which comprise most of the organic bone matrix. Additionally, it is able to activate (by cleaving) matrix-metalloproteinase-9 (MMP-9), which contributes to bone matrix degradation as well [41].

Significantly higher cathepsin K and MMP-9 immunoreactivity were found in the n-HA/ß-TCP composite group. However, no difference was found in the NFκ-β immunoreactivity nor the *RAMKL* RT-qPCR between the groups. These data suggest higher bone matrix turnover in the n-HA/ß-TCP composite group [42].

Transforming growth factor-β (TGF-β) is a superfamily comprising bone morphogenetic proteins (BMP) and TGF-β1, which are known to act locally on bone formation by stimulating the proliferation and chondrogenic or osteogenic differentiation of mesenchymal stem cells (MSCs). However, the TGF-β family is able to induce osteoclastogenesis and matrix resorption activity. Additionally, it can sustain mature osteoclast survival directly by way of osteoblast-osteocyte modulation. In this respect, the TGF-β superfamily has a central role in orchestrating bone matrix formation and resorption in bone remodeling [22,23,24,25,26,27,28,29,30,31,32,33,34,35,36,37,38,39,40,41,42,43]. We found higher TGF-β-immunireactivity in the n-HA/ß-TCP composite group, suggesting a higher rate of remodeling in this group.

In addition to matrix-metalloproteinase-9 (MMP-9), matrix-metalloproteinase-2 (MMP-2) is an important player in organic bone matrix resorption. However, some distinct characteristics are unique to MMP-2; it is related to osteocyte survival, canalicular maintenance, and osteogenic differentiation, in addition to osteoblast and osteoclast survival [44,45]. We found a greater MMP-2-immunoreaction in the n-HA/ß-TCP composite group, suggesting more sustained bone remodeling than that in the autogenous bone group.

Vascular endothelial growth factor (VEGF) is one of the most powerful inducers of angiogenesis, and it plays critical roles in bone development and remodeling [46]. Our data indicate higher VEGF immunoreactivity in the n-HA/ß-TCP composite group, suggesting enhanced potential for bone remodeling.

RUNX2 and OSX are the transcription factors more important to the induction of mesenchymal stem cells (MSCs) to differentiate toward the osteogenic lineage [22,23,24,25,26,27,28,29,30,31,32,33,34,35,36,37,38,39,40,41,42,43,44,45,46,47,48]. Our RT-qPCR data indicate higher RNAm expression of RUNX2 and OSX in addition to *ALKALINE PHOSPHATASE* in the autogenous bone group.

Osteocalcin (OCN) is a non-collagenous protein secreted by osteoblasts which can be stored in the bone matrix due to its capability to bind to hydroxyapatite. OCN can be activated by two different processes: via osteoclasts or an acidic pH. It is believed that OCN inhibits bone formation and induces hydroxyapatite crystals aligned parallel to the collagen bundles [49,50]. Our data indicate higher OCN gene expression in the autogenous bone group.

Figure 30 summarizes the hypothetical bone remodeling mechanisms from the data found in this work.

The overall panorama achieved in this experiment indicates that the n-HA/ß-TCP composite group presented significant gains in most of the substances tested, including traditional staining methods. On the other hand, most of the tested RNAm was higher in the autogenous bone group. It is worth remarking that RNAm’s presence is a prediction of near-future cellular events if transduction effectively takes place [51].

Synthetic biomaterials’ design, production, and testing is a fast-growing and promising field of research. As technologies evolve, the closer biomaterials properties become to native bone, eventually mimicking it and maybe surpassing it. If corroborated and extended by future experiments and by other research groups, the data presented in this work could be a relevant mark in synthetic biomaterial research.

## 4. Material and Methods

Twelve adult male *Wistar* rats 200–220 g in weight, provided by the Roberto Alcantara Biology Institute of Rio de Janeiro State University, were used. The animals were kept in individual cages with ad libitum access to food and water. The light/dark cycle (lights on at 7:00 a.m. and off at 7:00 p.m.) and temperature (22 °C) were kept constant. The experimental animal procedures protocol was approved by the local animal ethics committee (#001/2019).

The rats were anesthetized with ketamine hydrochloride/xilasine solution (1/1, 0.1 mg/kg, i.p.). The dorsal cranium was trichotomized, and a sagittal incision was carried out using a sterile surgical scalp. Skin and periosteum were cleared in the bilateral parietal region to prepare the bone defect. The bone defect was made bilaterally using a sterilized punch (cutting edge Ø: 3 mm). The bone fragments were carefully removed to avoid damage to the duramater and related blood vessels. After biomaterial insertion into the bone defect, the skin was carefully replaced and cotton wire sutured.

The rats were distributed into 2 groups:

Group 1 (*n* = 6): The bone defects were filled with 0.1 g of calvarias autogenous bone (size: 0.5–1 mm).

Group 2 (*n* = 6): The bone defects were filled with 0.1 g of the n-HA/ß-TCP composite (size: 0.5–1 mm).

Sixty days after surgery, the animals were euthanized under deep anesthesia using ketamine hydrochloride/xilasine solution (1/1, 0.3 mg/kg, i.p.). The animals were decapitated, and the heads were sent to histological procedures.

### 4.1. X-Ray Diffraction

X-ray diffraction (XRD) was used to analyze the n-HA/ß-TCP composite. A Panalytical Empyrean (Almelo, Netherlands) diffractometer was used, with Cu-Kα radiation, a 2θ range of 20°–80°, a step width of 0.02°, and 5 s of exposure time. Identification of the diffraction peaks was based on comparisons with International Center for Diffraction Data (IDCC) diffraction files and the COD-Jan2012 (Crystallography Open Database) and PDF2-2004 databases. XRD data were analyzed using the Rietveld method to identify and quantify the phase percentage.

### 4.2. Morphological Analysis Protocol

The animal heads were trimmed and fixed in 4% paraformaldehyde for 48 h and decalcified in 10% EDTA (pH = 7.0) for 6 weeks. The samples were dehydrated in a series of graded solutions (70%, 90%, and 100% ethanol) and clarified in xylol, followed by paraffin embedding (Paraplast, Sigma-Aldrich, St. Louis, MO, USA) at 65 °C. Serial sections 7 µm thick were cut using a microtome (LEICA, Nussloch, Germany), and collected in silanized slides.

Histomorphometry evaluation was performed using three staining methods (Goldner trichrome, PAS, and Sirius red) and TRAP histochemistry.

The slides were deparaffinized in xylene, rehydrated with alcohol (100%, 95%, and 70%), washed in distilled water, and immersed in Weigert’s iron hematoxylin (Sigma-Aldrich, St. Louis, MO, USA) for 10 min. After washing in distilled water, the slides were immersed in a Biebrich scarlet-acid fuchsin solution (Sigma-Aldrich, St. Louis, MO, USA) for 15 min, washed again with distilled water, and immersed in a phosphomolybdic-phosphotungstic acid solution for 10 min. This was followed by aniline blue solution immersion for 5 min before they were washed again with distilled water, dehydrated, and cover slipped.

The slides were deparaffinized and rehydrated with alcohol (100%, 95%, and 70%), washed in distilled water, immersed in periodic acid (1%) for 5 min, and rinsed in distilled water. The slides were then immersed in Schiff’s reagent (Fuchsin Basic (1%), sodium metabisulphite (2%) in HCl (2%) solution) for 5–15 min, followed by being wash in running tap water for 5–10 min and counterstained with Herri’s hematoxylin for 15 s.

The slides were deparaffinized and bathed in acid phosphatase Burstone solution. The Burstone solution was a mixture of naphthol AS-BI phosphate substrate (Sigma-Aldrich, St. Louis, MO, USA) (4 mg) diluted in 0.25 mL of N–dimetil-formamida (Sigma-Aldrich, St. Louis, MO, USA) and 25 mL of acetate buffer (0.2 M pH 5.0), 35 mg of “Fast Red Violet” LB Salt (Sigma-Aldrich, St. Louis, MO, USA), and 2 drops of MgCl (10%), followed by a slide bath for 3 h in D(−) tartaric acid (0.2352 g) (Sigma-Aldrich, St. Louis, MO, USA). After this bath, the slides were rinsed in tap water, counterstained with Harris hematoxilin (10%), clarified in xylol, and cover slipped in Entellan (EMS, Hatfield, PA, USA).

The histological slides were initially deparaffinized in xylol baths (3 × 5 min) and hydrated in decreasing concentrations of alcohol (100%, 90%, and 70%) for 5 min in each bath. They were then incubated in 3% hydrogen peroxide solution for 15 min in the dark to inhibit endogenous peroxidase. After the inhibition of endogenous peroxidase activity, rinsing in PBS buffer at a pH of 7.2 was performed (3 × 5 min).

Antigenic site re-exposure was conducted in citrate buffer solution (pH of 6.0 at 96 °C for 20 min). After the slides cooled and were rinsed with a PBS buffer at a pH of 7.2 (3 × 5 min), nonspecific sites were blocked with PBS/BSA solution (3% for 20 min). After rinsing, the slides were incubated with the primary anti-VEGF antibody (Santa Cruz, sc-1876), diluted in PBS/BSA 1% (1:100) primary anti-OPN, anti-CAL, anti-TRAP, anti-acid phosphatase, anti-VEGF, anti-TNF-alfa, anti-BMP-2, anti-BMP-9, and anti-TGF-B antibody (Santa Cruz, sc-21742) and diluted in PBS/BSA 1% (1:200) overnight in a refrigerator (4.0 °C) in a humid chamber. After primary antibody incubation, 3 baths in PBS buffer solution at a pH of 7.2 (5 min) were carried out, followed by secondary biotinylated antibody (VECTASTAIN Universal Quick HRP Kit, Ingold Road, Burlingame, CA, USA) incubation for 1 h at room temperature. After another PBS buffer solution rinse, the slides were incubated with streptavidin (VECTASTAIN Universal Quick HRP Kit) for 30 min at room temperature. The streptavidin–biotin–peroxidase complex was revealed with diaminobenzidine (DAB) (VECTASTAIN Universal Quick HRP Kit). The slides were counterstained with hematoxylin solution (0.15%), dehydrated in increased alcohol concentrations (70%, 90%, and 100%) (ethanol), diaphanized, and mounted using Entellan resin (Sigma-Aldrich, St. Louis, MO, USA).

### 4.3. Image Acquisition and Histomorphometry

The slices stained with Goldner’s Masson trichrome, Schiff’s periodic acid, or immunostaining were observed under a light microscope equipped with a CCD camera (Olympus BX53 with Olympus DP72 camera; Nagano, Chubu, Japan) at a 400× magnification. The quantification of each stain (three random fields) was performed with Image-Pro Plus 7.0 (Media Cybernetics, Silver Springs, Rockville, MD, USA) (Figure 31) and Graph Pad é programa estatístico (já está descirto no tópico análise estatística).

Three randomized slide fields in Goldner trichrome, PAS staining and immune labeling were photographed with a photomicroscope (Carl Zeiss—JVC TK-1270 color video camera) at a 200× magnification. The images were quantified using GraphPad Prism Version 8.0 (Figure 31). The Figure 32 shows the low-magnification of the bone defect evaluated.

### 4.4. RNA Isolation, Reverse Transcription, and Quantitative Real-Time PCR (qRT-PCR) Analysis

The parietal bone of 6 rats per group was removed before fixation procedures, imbibed in RNA later with a stabilization solution (Invitrogen, Cat. AM7021, Carlsbad, CA, USA), and frozen at −80 °C until analysis. To isolate and purify the total RNA, a PureLink RNA Mini Kit (Invitrogen, Cat. 12183018A, Carlsbad, CA, USA) was used for datasheet information. The total RNA was quantified by Qubit using Quant It Kit RNA (Invitrogen, Cat. Q32852, Carlsbad, CA, USA), followed by DNAse treatment and a new total RNA quantification. For the reverse transcriptase, a concentration of 500 ng/mL total RNA was used, followed by being submitted to a SuperScript III First-Strand Synthesis System with Oligo dT prime (Invitrogen, Cat. 18080051, Carlsbad, CA, USA) in a cycle of 65 °C/5 min; 50 °C/50 min; 85 °C/5 min; and 37 °C/20 min, as described in the datasheet. The cDNA product was diluted to a final concentration of 50 ng/mL with RNAse-free water, and 2 μL of this solution was used for qPCR. For amplification, the Forget-Me-Not EvaGreen qPCR Master Mix (Biotium, Cat. 31046, Fremont, CA, USA) was used in 1 cycle of 95 °C/2 min followed by 40 cycles of 95 °C/5 s; 55 °C/10 s; and 72 °C/20 s, as described in the datasheet.

The primer sequences are listed in Table 1. The dissociation melting curve was formed according to instrument guidelines, and the result was expressed using β-actin as a housekeeping gene and 2-DDCt to calculate the relative fold expression.

### 4.5. ELISA Validation Test for Analysis of Acid Phosphatase (AP) Activity in Bone

The blood plasma and bone of 5 control animals were collected and used to compare the acid phosphatase activity. To extract AP from the bone samples, we homogenized 6–10 mg of the bone tissue with 0.5 mL of lysis buffer (0.3 M KCl, 0.1% Triton X-100). The samples were centrifuged at 20.744 rpm for 20 min at 4 °C, and the supernatant was used for kit validation. Validation of the AP activity in the bone samples was realized with an Acid Phosphatase Colorimetric Assay Kit (#10008051, Cayman Chemical (Michigan), Ann Arbor, MI, USA) and followed the datasheet method.

Briefly, 20 mL of each sample was added into a 96 well plate with and without tartrate inhibitor followed by the acid phosphatase substrate. The plate was incubated for 20 min at 37 °C followed by a stop solution. The absorbance was measured at 405 nm, and the AP activity was calculated as suggested in the kit datasheet.

### 4.6. Acid Phosphatase Assay

The bone samples were homogenized in 0.5 mL of lysis buffer (0.3 M KCl, 0.1% Triton X-100). The samples were centrifuged at 20.744 rpm for 20 min at 4 °C, and the supernatant was used for total protein quantification (Pierce™ BCA Protein Assay Kit, #23225, ThermoScientific (Carlsbad), Carlsbad, CA, USA) and the AP assay (Acid Phosphatase Colorimetric Assay Kit, #10008051, Cayman Chemical, Ann Arbor, MI, USA). About 7.6 mg of total protein was used to quantify the AP activity in the bone as described in the datasheet.

Each sample was added into the 96 well plate with and without tartrate inhibitor followed by the acid phosphatase substrate. The plate was incubated for 20 min at 37 °C followed by a stop solution. The absorbance was measured at 405 nm, and the AP activity was calculated as suggested in the kit datasheet.

Data were analyzed using one-way ANOVA followed by a Kruskal–Wallis test (*p* < 0.05). All analyses were performed using the GraphPad Prism Version 8.0 and BioEstat 5.0 software.

### 4.7. Scanning Electron Microscopy

Immediately after euthanasia, samples were collected from the surgery sites. The samples were cut into small tissue blocks (1 mm^3^) and fixed in 2.5% glutaldehyde in sodium cacodylate buffer (0.1 M) at a pH of 7.4 at 4 °C for 12 h. After being washed in 0.1 M sodium cacodylate buffer, fragments were postfixed with 1% osmium tetroxide diluted in 0.1 M sodium cacodylate buffer. After further washing, the material was dehydrated in an increasing series of ethanol (30%, 50%, 70%, 90%, and 2× absolute) for 30 min at each step. The material was taken to the critical point device (Critical Point Dryer—CPD 030, Bal-Tec, Amstetten, Germany) for the replacement of ethanol by CO_2_ and later fixed in stubs with carbon tape and metalized with gold.

Before SEM analysis, the samples were coated with gold film. An SEM Quanta 250 FEV (Thermo Fischer Scientific, Waltham, MA, USA) was used. The magnifications used were 5000× to evaluate biomaterial homogeneity, 15,000× to evaluate cell clusters and architecture, and 20,000× to evaluate specific cell types.

## 5. Conclusions

Based on the presented results, we can affirm that the nanohydroxyapatite influenced more effective osteoinduction than the autogenous bone, indicating the following:Nanogeometry creates a more favorable structure for cell adhesion and development.There was an intense formation of bone matrix in Goldener’s trichrome staining (PAS and PSR).In the immunohistochemistry, there was a prevalence of cells such as osteoblasts and osteoclasts for nanohydroxyapatite, in addition to several potentially more abundant blood vessels, when analyzed with VEGF.The rt-PCR test and Elisa test also confirmed the results shown in the previous tests, demonstrating a significantly more favorable response for the bone maturation process in the case of the nanohydroxyapatite/beta-tricalcium phosphate composite.In the ultrastructural analysis, it was possible to describe that in the same period, the bone physiology was better defined and structured in the nanohydroxyapatite than in the autogenous bone.

## 6. Future Perspectives

In vitro analysis should provide additional biochemical data for a more detailed understanding of the naïve bone–biomaterial interactions and possibly improve biomaterial designs further.

## Figures and Tables

**Figure 1 ijms-26-00052-f001:**
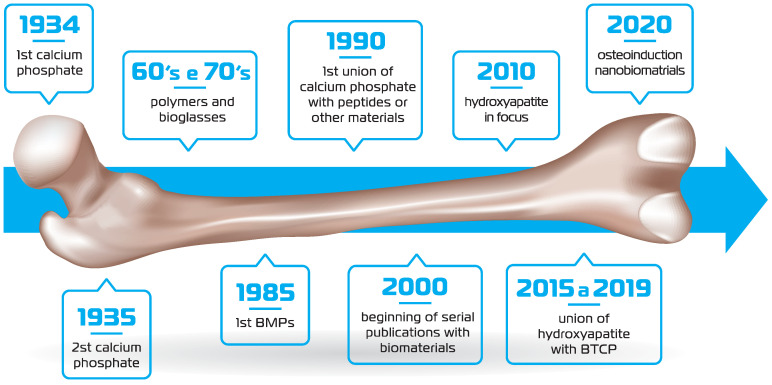
Timeline of the main events related to biomaterial design and research.

**Figure 2 ijms-26-00052-f002:**
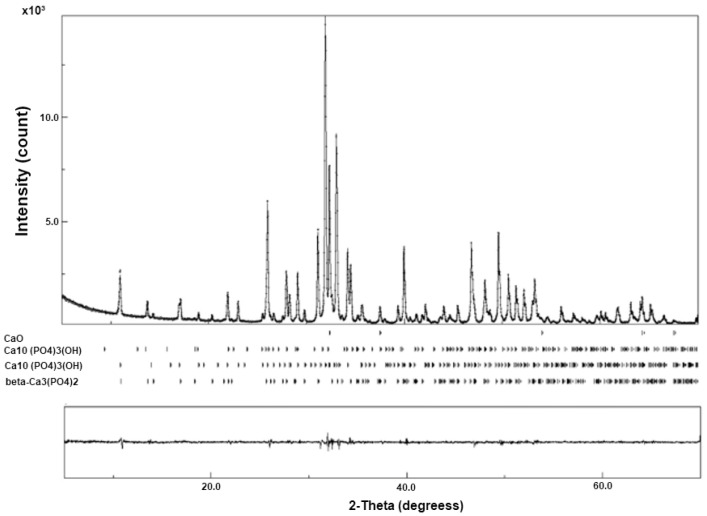
Diffractogram of the n-HA/ß-TCP compound. The peaks show the presence of three phases.

**Figure 3 ijms-26-00052-f003:**
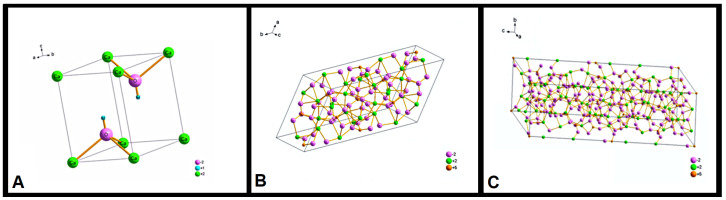
Crystal structure of the n-HA/ß-TCP compound. (**A**) β-Ca_3_(PO_4_)_2_ with trigonal symmetry. (**B**) Ca_10_(PO_4_)_3_(OH) with hexagonal symmetry. (**C**) Ca_10_(PO_4_)_3_(OH) with monoclinic symmetry.

**Figure 4 ijms-26-00052-f004:**
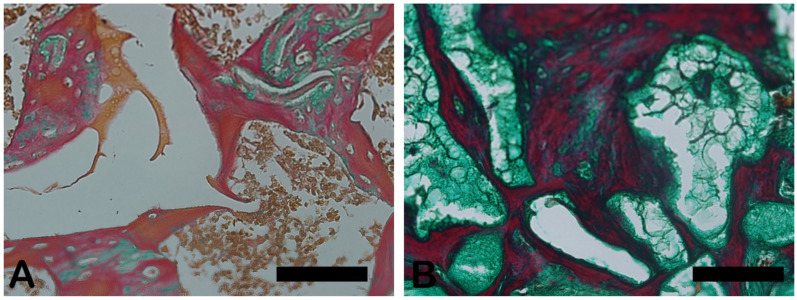
Goldner’s trichrome-stained slide photomicrographs. The osteoid bone matrix presents a red color, and the bone matrix mature collagen can be observed in green. (**A**) Autogenous group. (**B**) Blue Bone group (REG). Scale bar = 100 µm, ×200 magnification.

**Figure 5 ijms-26-00052-f005:**
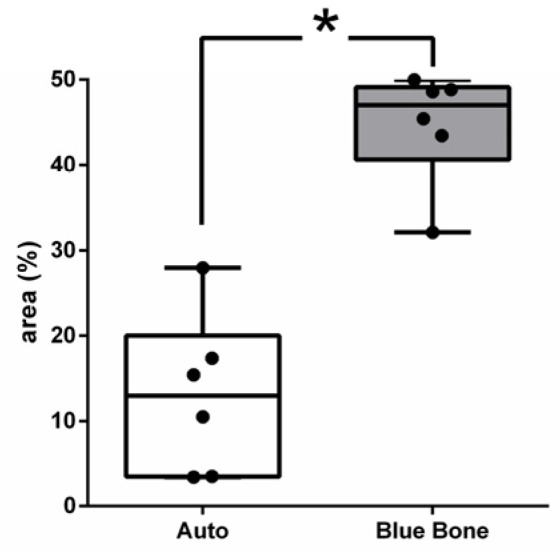
Result of the histomorphometric analysis in the osteoid bone matrix bone areas stained by Goldner’s trichrome method (autogenous group and n-HA/ß-TCP compound). * *p* = 0.0022.

**Figure 6 ijms-26-00052-f006:**
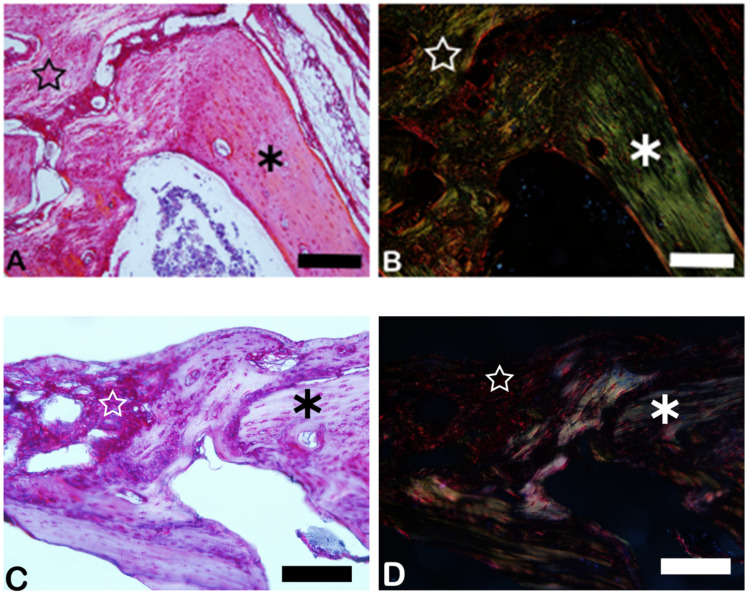
Picrosirius red-stained sections, observed under non-polarized (**A**,**C**) and polarized light (**B**,**D**), of the critical defects to detect collagen during bone remodeling. Polarized light can be used to distinguish the maturation (thickness and packing) of type 1 collagen in bone tissue. Yellow and red stains represent mature type 1 collagen, and green represents immature type 1 collagen. Bone calvaria represented by asterisks. Critical defects represented by stars. Scale bar = 100 µm, ×200 magnification.

**Figure 7 ijms-26-00052-f007:**
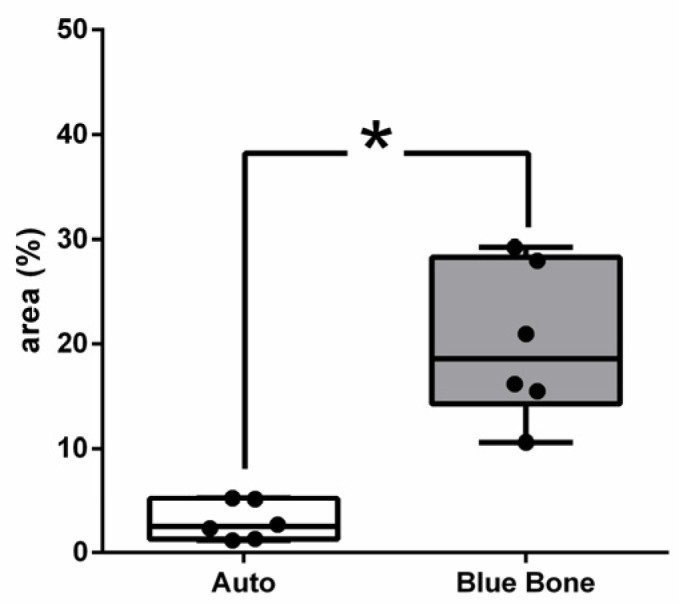
Result of the histomorphometric analysis in the osteoid bone matrix bone and collagenous areas stained by picrosirius red (autogenous group and n-HA/β-TCP compound). * *p* = 0.0022.

**Figure 8 ijms-26-00052-f008:**
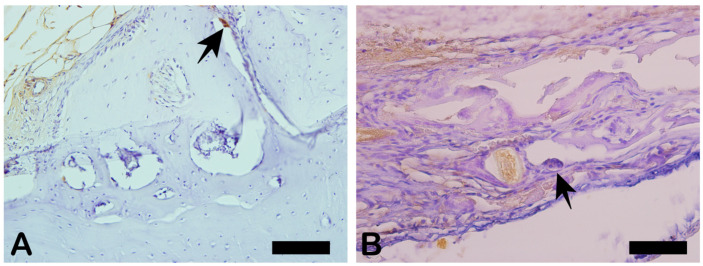
Photomicrographs of cathepsin K immunostaining (dark brown color) in autogenous bone (**A**) and n-HA/β-TCP composite (**B**). Black arrows indicate osteoclasts. Scale bar = 100 µm, ×200 magnification.

**Figure 9 ijms-26-00052-f009:**
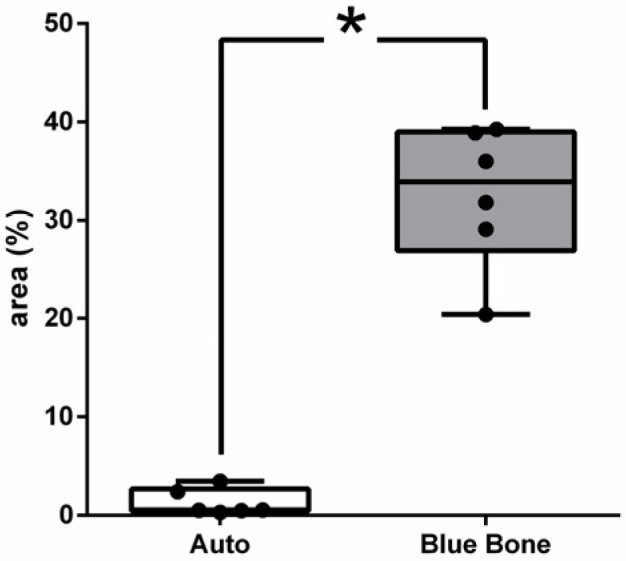
Result of the histomorphometric analysis of osteoclast regulation through osteoblast activity by cathepsin K (autogenous bone group and n-HA/β-TCP composite group). * *p* = 0.0022.

**Figure 10 ijms-26-00052-f010:**
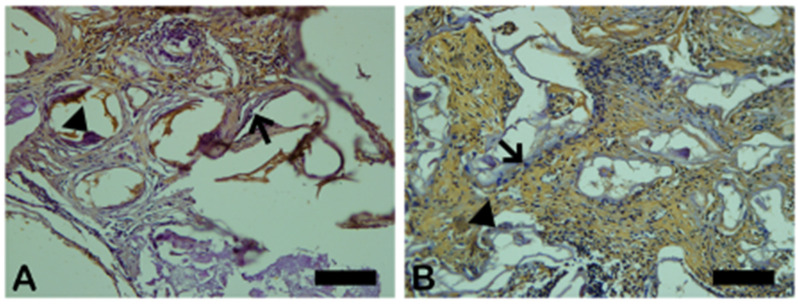
Immunohistochemical labeling for TGF-B. (**A**) Moderate marking in osteoclasts (arrowheads), osteoblasts (arrows), and mesenchymal cells (asterisks). Slight staining of newly formed bone and intense staining appeared in extracellular matrix (star). (**B**) Slight expression in osteoclasts (arrowheads), osteoblasts (arrows), and more intense expression in mesenchymal cells (asterisks). Intense and diffuse staining in extracellular matrix. Scale bar = 100 µm, ×200 magnification.

**Figure 11 ijms-26-00052-f011:**
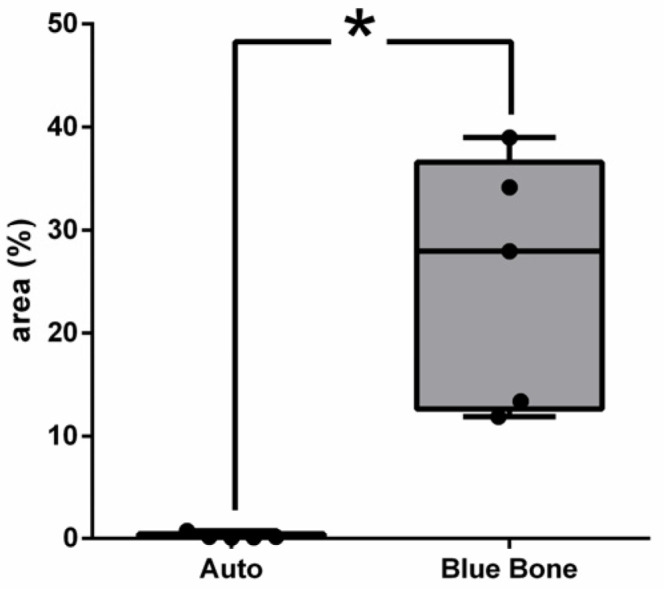
Result of the histomorphometric analysis of immunostaining to TGF-β (autogenous bone group and n-HA/β-TCP composite group). * *p* = 0.0079.

**Figure 12 ijms-26-00052-f012:**
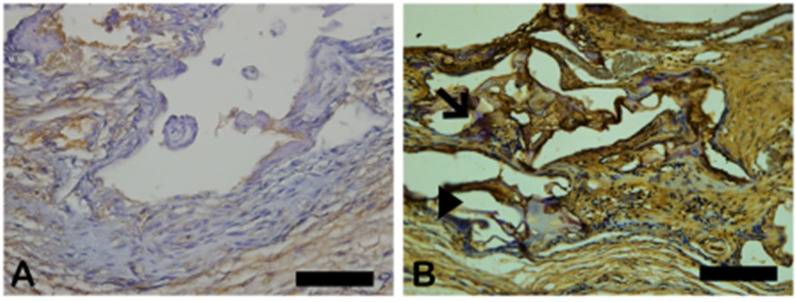
Immunohistochemical labeling for OPN. (**A**) Extracellular OPN appeared to be extremely low in extracellular matrix. (**B**) Intense and diffuse staining in extracellular matrix. The arrow indicates new-formed trabeculae and arrowhead indicates intertrabecular space. Scale bar = 100 µm, ×200 magnification.

**Figure 13 ijms-26-00052-f013:**
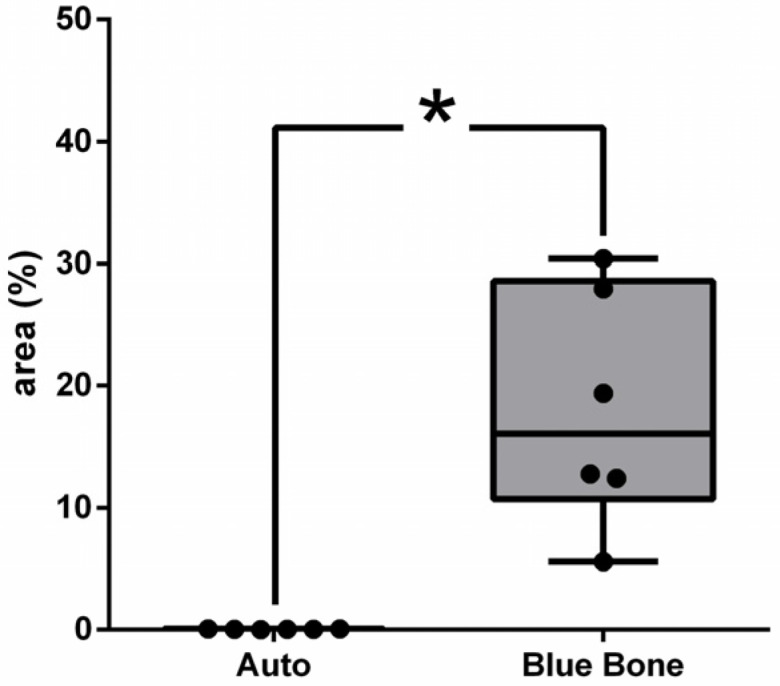
Result of the histomorphometric analysis of immunostaining for OPN (autogenous bone group and n-HA/β-TCP composite group). * *p* = 0.0022.

**Figure 14 ijms-26-00052-f014:**
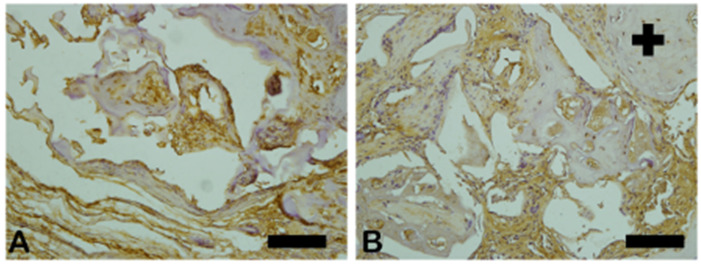
VEGF-stained photomicrographs. The brownish deposits indicate VEGF’s presence and its activity, which is involved in the process of formation of new blood vessels. (**A**) Autogenous bone group. (**B**) n-HA/β-TCP group. Cross sign indicates mineralized bone matrix. Scale bar = 100 µm, ×200 magnification.

**Figure 15 ijms-26-00052-f015:**
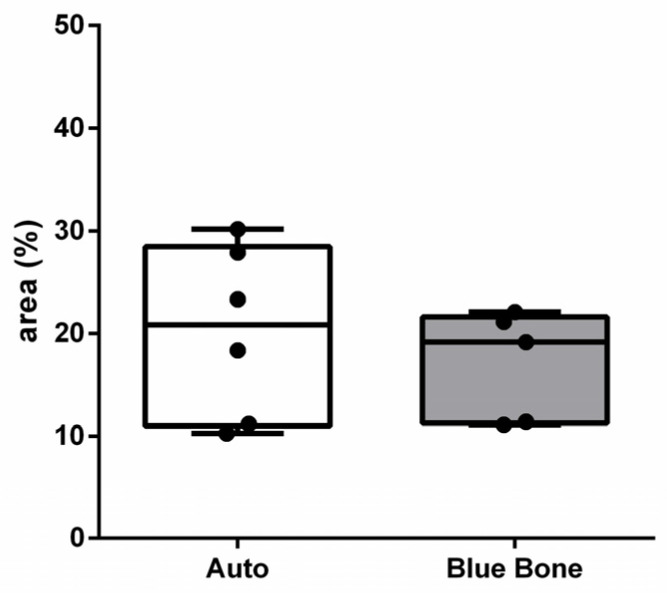
Result of the histomorphometric analysis of immunostaining for VEGF (autogenous bone group and n-HA/β-TCP composite group). *p* = 0.6277.

**Figure 16 ijms-26-00052-f016:**
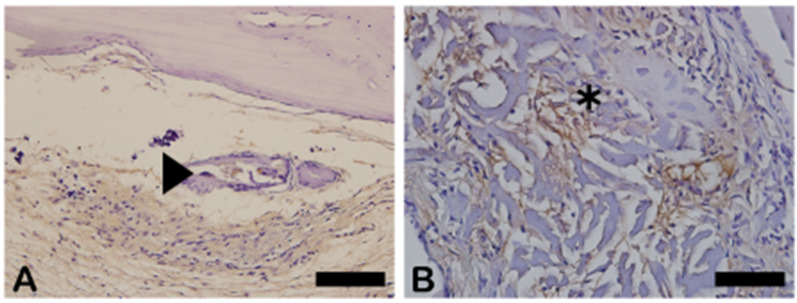
Immunohistochemical labeling for nuclear factor kappa B (NFKB). (**A**) Expression of NFKB was observed slightly in extracellular matrix and osteoclasts (arrowhead). (**B**) Slightly more intense staining was observed in extracellular matrix and fibroblast-like cells (asterisk).

**Figure 17 ijms-26-00052-f017:**
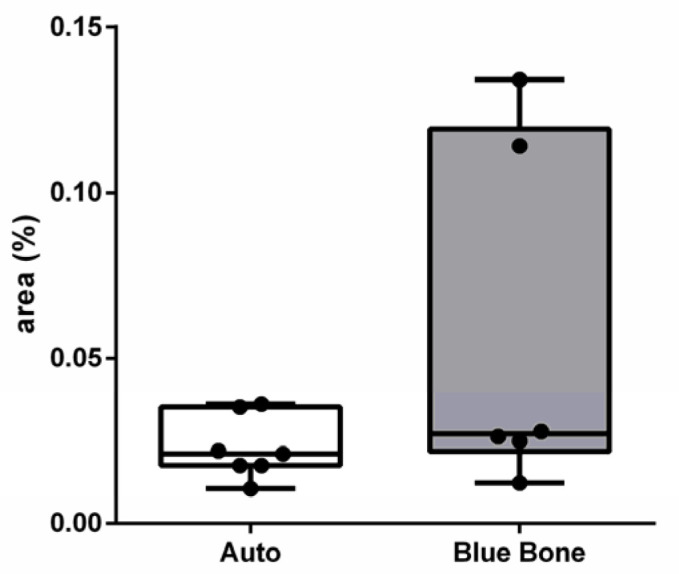
Result of the histomorphometric analysis of immunostaining for NFκ-β (autogenous bone group and n-HA/β-TCP composite group). *p* = 0.2191.

**Figure 18 ijms-26-00052-f018:**
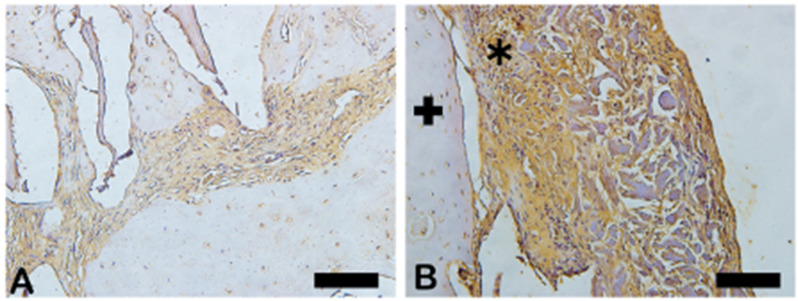
Immunohistochemical labeling for metaloproteinases-9. (**A**) Expression of MMP-9 observed slightly in extracellular matrix. (**B**) Strong staining marks observed in extracellular matrix (asterisk). In both groups, osteocytes (cross) and fibroblast-like cells were also immunostained, being more intense in group B. Scale bar = 100 µm, ×200 magnification.

**Figure 19 ijms-26-00052-f019:**
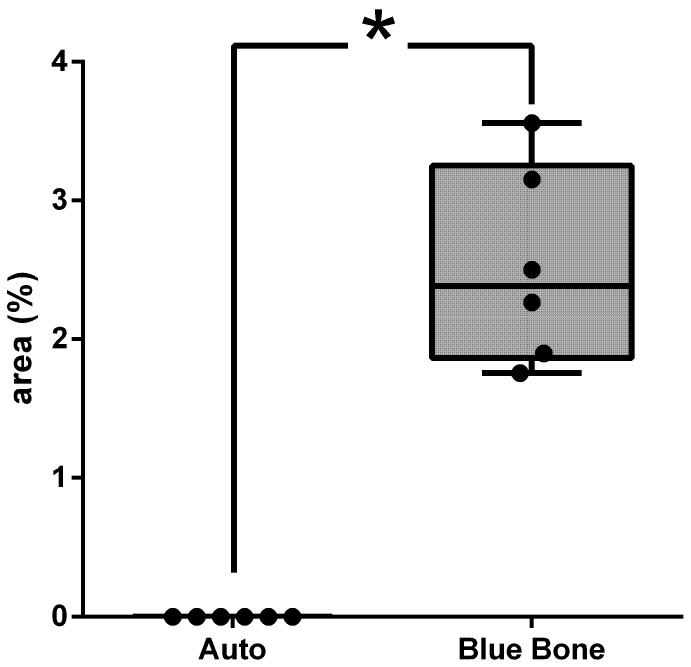
Result of the histomorphometry analysis of immunostaining for MMP-9 (autogenous bone group and n-HA/β-TCP composite group). * *p* = 0.0022.

**Figure 20 ijms-26-00052-f020:**
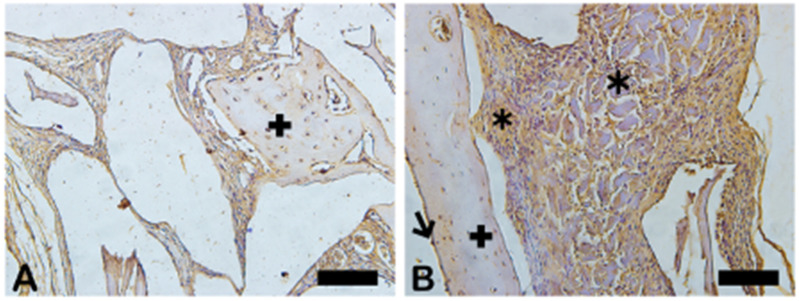
Immunohistochemical labeling for metalloproteinase-2. (**A**) Expression of MMP-2 was observed slightly in extracellular matrix and in several osteocytes (cross). (**B**) Strong marking was observed in extracellular matrix, osteoblasts (arrow), osteocytes (cross), and fibroblast-like cells (asterisks). Scale bar = 100 µm, ×200 magnification.

**Figure 21 ijms-26-00052-f021:**
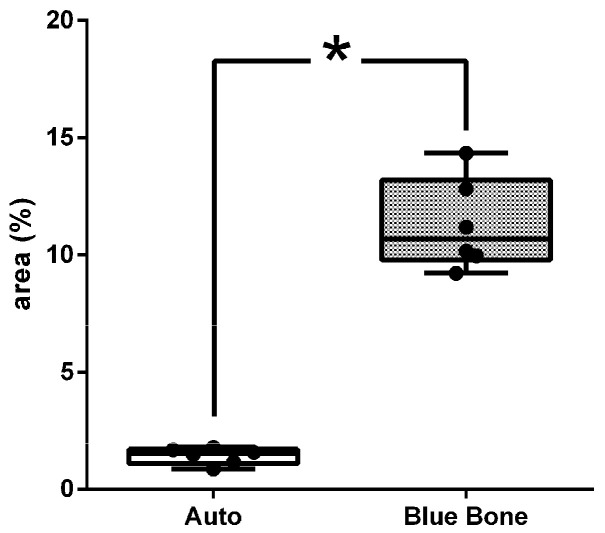
Result of the histomorphometric analysis of immunostaining for MMP-2 (autogenous bone group and n-HA/β-TCP composite group). * *p* = 0.0025.

**Figure 22 ijms-26-00052-f022:**
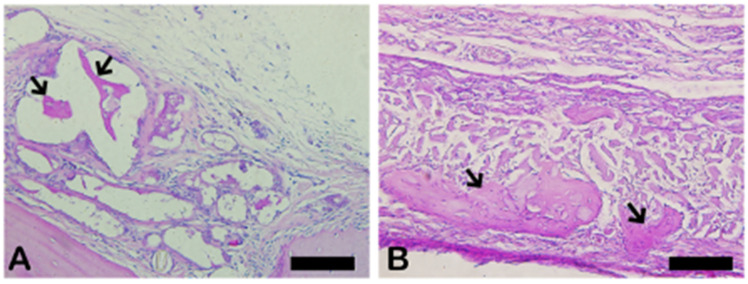
Photomicrographs of critical defect slides stained with Schiff’s periodic acid (PAS), showing a positive reaction for calcified bone (pink) and a negative reaction for osteoid tissue. (**A**) The black arrows show the thin trabeculae of mineralized bone observed in the autogenous group. (**B**) The black arrows show the thick trabeculae of mineralized bone observed in the Blue Bone group (REG). Scale bar = 100 µm, ×200 magnification.

**Figure 23 ijms-26-00052-f023:**
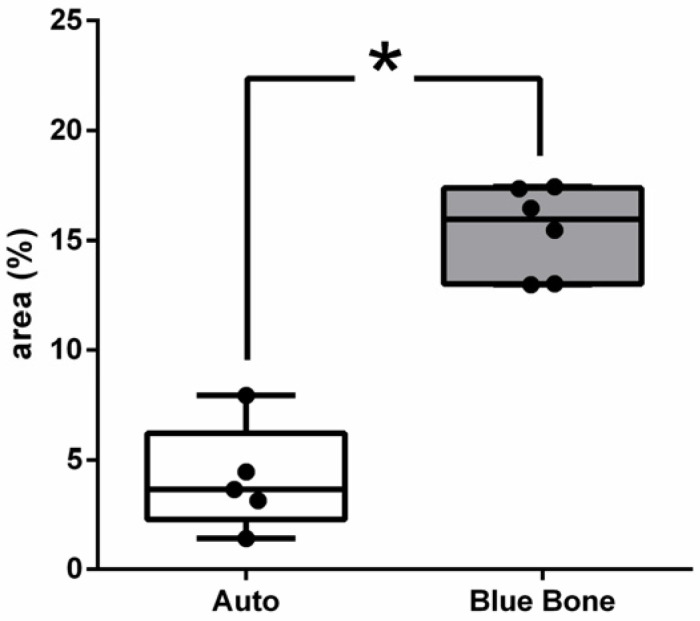
Result of the histomorphometric analysis of PAS histochemistry (autogenous bone group and n-HA/β-TCP composite group). * *p* = 0.0043.

**Figure 24 ijms-26-00052-f024:**
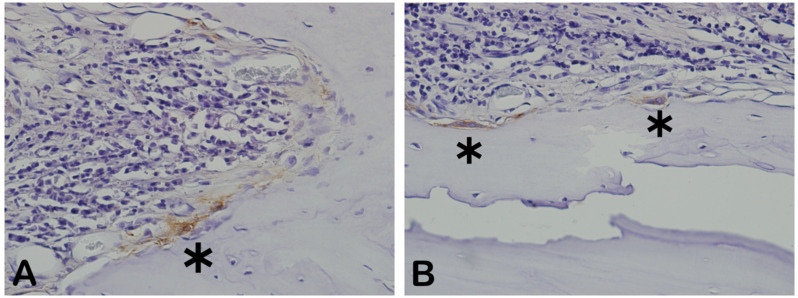
Photomicrographs of metalloproteinase (BMP-2)-stained slides. Brown color indicates osteoblastic activity. (**A**) Group 1 (Blue Bone). (**B**) Group 2 (autogenous). Scale bar = 100 µm, 20× magnification.

**Figure 25 ijms-26-00052-f025:**
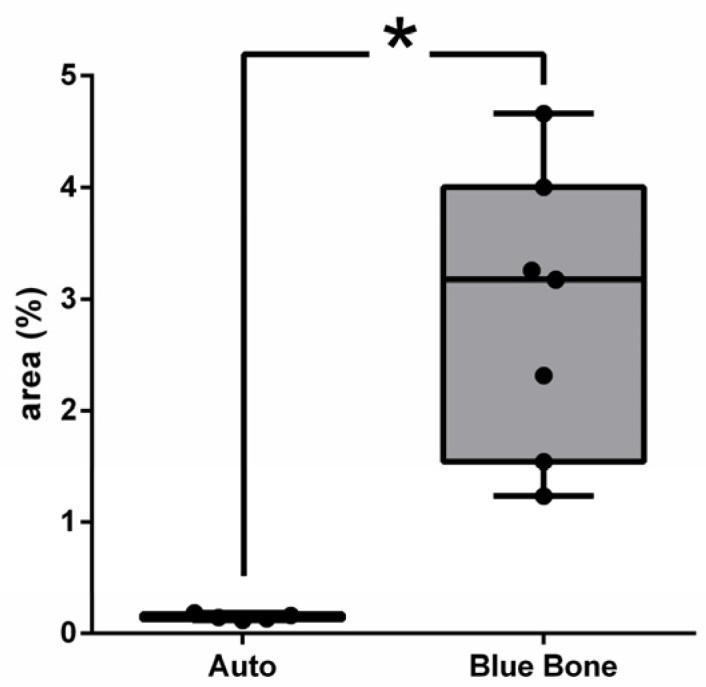
Result of the histomorphometric analysis of inflammation regulation through osteoblast activity by BMP-2 (autogenous group and Blue Bone group (REG)). * *p* < 0.0001.

**Figure 26 ijms-26-00052-f026:**
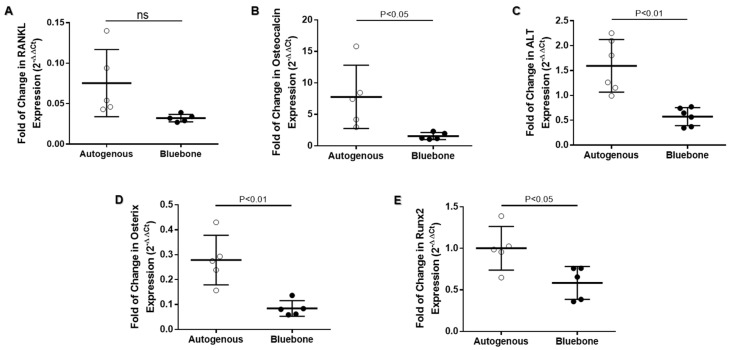
Expression analysis of *RANKL* (**A**), *OSTEOCALCIN* (**B**), *ALP* (**C**), *OSTERIX* (**D**), and *RUNX2* (**E**) in calvaria of rats with two types of bone grafts (autogenous and Blue Bone™) 60 days after induced injury. The results represent the fold chance of each mRNA expression upon the control, expressed as the mean of 5 animals per group ±SD. For statistical analysis, the values were submitted to a parametric unpaired *t*-test with Welch’s correction, and *p* < 0.05 between groups was considered significant. Legend: RANKL = receptor activator of nuclear factor-κB ligand; ALP = alkaline phosphatase; Runx2 = Runt-related transcription factor 2.

**Figure 27 ijms-26-00052-f027:**
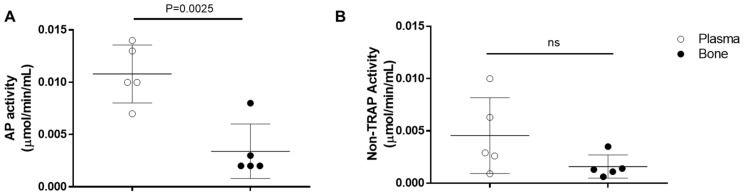
Validation of bone-derived acid phosphatase activity. To validate the Acid Phosphatase Colorimetric Assay Kit (#10008051, Cayman Chemical, Ann Arbor, MI, USA), we used 20 μL of plasma and bone-extracted supernatant as described in the datasheet. (**A**) Analysis of total acid phosphatase activity. (**B**) Analysis of non-tartrate-resistant acid phosphatase activity. Groups are represented by mean ± SD. Statistics were derived via one-way ANOVA with Holm–Sidak posttest. ns: no significant.

**Figure 28 ijms-26-00052-f028:**
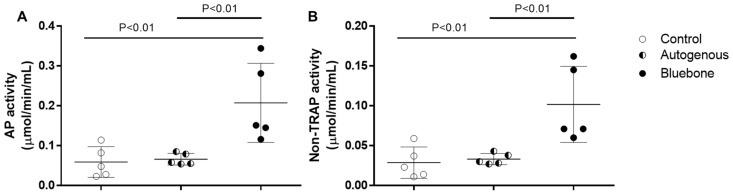
Analysis of bone-derived acid phosphatase (AP) activity in calvaria of rats grafted with autogenous bone or Blue Bone. (**A**) Analysis of total acid phosphatase activity. (**B**) Analysis of non-tartrate-resistant acid phosphatase activity. Groups are represented by mean ± SD. Statistics accessed via one-way ANOVA with Holm–Sidak posttest.

**Figure 29 ijms-26-00052-f029:**
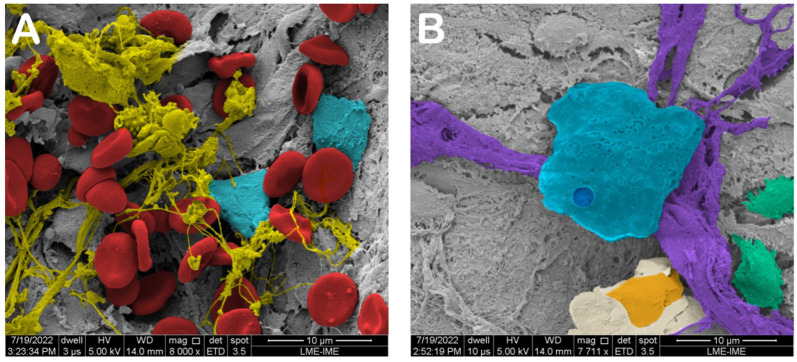
Scanning electron micrographs. (**A**) The autogenous graft group, showing red blood cells (red), collagen fibrils (yellow), and endothelial cells (light blue) at ×8000 magnification. (**B**) The n-HA/β-TCP composite group, showing the biomaterial particles (light blue), collagen fibers (purple), osteoblasts (orange), and osteocytes inside lacunae (green) at ×7711 magnification.

**Figure 30 ijms-26-00052-f030:**
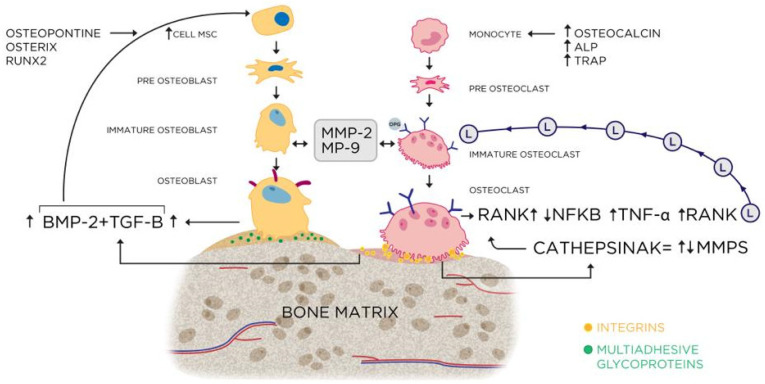
Representative scheme of the activation cascade of the bone remodeling process involving the markers present in this study.

**Figure 31 ijms-26-00052-f031:**
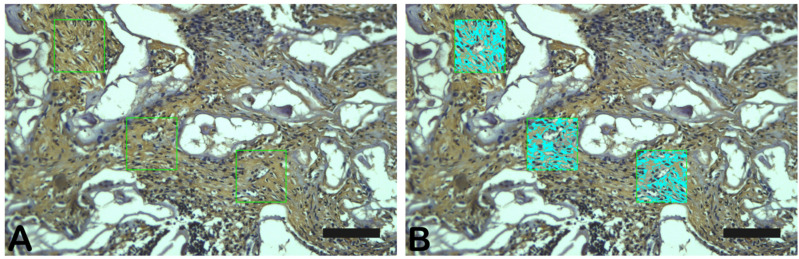
Photomicrograph of TGF-β immunostaining as an example of data collection. (**A**) choice of area, (**B**) data collection. Scale bar = 100 µm, 200× magnification.

**Figure 32 ijms-26-00052-f032:**
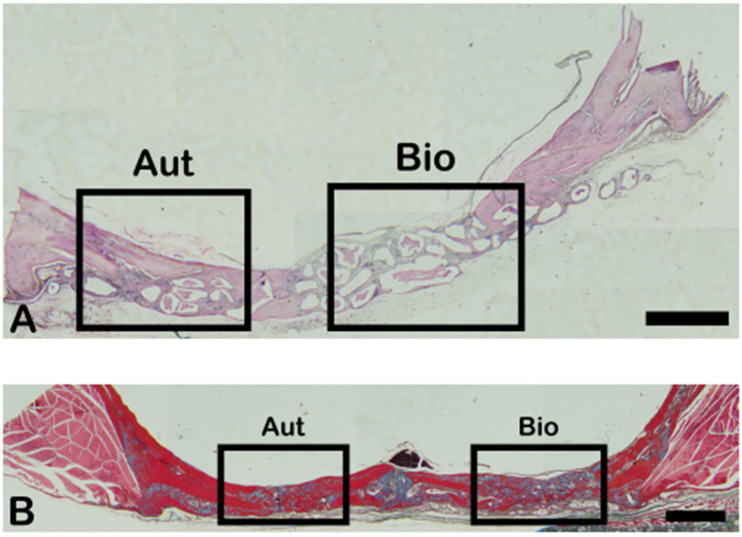
Photomicrograph of Goldner’s Masson trichrome (**B**) and hematoxin and eosin (**A**) coloring as an example of data collection. Scale bar = 100 µm, 1–25× magnification.

**Table 1 ijms-26-00052-t001:** Primer sequences.

Primer		Forward	Reverse
*β-actin*		5′-CAGAGCAAGAGAGGCATCCT-3′	5′-GTCATCTTTTCACGGTTGGC-3′
*RANKL*		5′-TCGCTCTGTTCCTGTACT-3′	5′-AGTGCTTCTGTGTCTTCG-3′
*Osteocalcin*		5′-CACAGGGAGGTGTGTGAG-3′	5′-TGTGCCGTCCATACTTTC-3′
*ALP*		5′-GCCTGGACCTCATCAGCATT-3′	5′-GGGAAGGGTCAGTCAGGTTG-3′
*Osterix*		5′-GCCTACTTACCCGTCTGA-3′	5′-CTCCAGTTGCCCACTATT-3′
*Runx2*		5′-TAACGGTCTTCACAAATCCTC-3′	5′-GGCGGTCAGAGAACAAACTA-3′

## Data Availability

All data generated or analyzed during this study are included in this published article.

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
