# Peer review of "Comparison Between Nano-Hydroxyapatite/Beta-Tricalcium Phosphate Composite and Autogenous Bone Graft in Bone Regeneration Applications: Biochemical Mechanisms and Morphological Analysis"

_ijms, 2024, doi:10.3390/ijms26010052_

Round 1
Reviewer 1 Report
Comments and Suggestions for Authors
There is no reference to the fixation procedures of the heads.
In section 2.4: RNA isolation, reverse transcription, and quantitative real-time PCR (qRT-PCR) analysis. there is a reference of another bone samples (frontal bone) obtained. From the same animals? In what moment? Must be specified
Author Response
Response to Reviewer Comments
Reviewer 1
- There is non reference of the fixation procedures of the reads.
Answer: A more adequate description of the fixation procedures was added in Material and Methods section, as suggested.
- In tha section 2.4: RNA isolation, reverse transcription and qualitative real-time PCR analysis there is reference of another bone samples (frontal bone) obtained from the same animals? At what moment? Must be specified.
Answer: Correction was done in the manuscript. The parietal bone was used in the experiments.
Reviewer 2 Report
Comments and Suggestions for Authors
This study compared bone regeneration using a nano-hydroxyapatite/β-tricalcium phosphate (n-HA/β-TCP) composite and autogenous bone grafts. The author compared the two materials in various ways after bone defects
1. It seems unnecessary to separate the immunostaining data and the related graphs.
2. The author needs to include additional content about the bone defect area by adding low-magnification photographs and staining with H&E and Masson's trichrome stain.
3. A three-dimensional evaluation using micro-CT is needed to demonstrate that n-HA/β-TCP is better for bone regeneration than autogenous bone grafts
4. An explanation is needed regarding the particle sizes of autogenous bone and n-HA/β-TCP implanted in the bone defect area
5. Please explain in detail whether n-HA/β-TCP is more efficient for bone regeneration than autogenous bone graft
Comments on the Quality of English LanguageThe author needs some wording and grammar corrections
Author Response
Response to Reviewer Comments
Reviewer 2
- It seems unnecessary to separate the immunostaining data and related graphs.
Answer: The reviewer’s suggestion is right, but the authors considered that the size of the photomicrographs would be too small, so we kept as it is.
- The author needs to include additional content about the bone defect área by adding low-magnification photographs and staining with H&E and Masson’s trichrome stain.
Answer: The authors considered the reviewer’s suggestion truly important, so we added a low-magnification H&E and PCS photmicrograph in the manuscript.
- A three dimensional evaluation using micro CT is needed to demonstrate that n-HA/bTCP is better for bone regeneration than autogenous bone graft.
Answer: The reviewer’s suggestion is perfectly right again, and it would make a more solid argument about n-HA/bTCP better for bone regeneration. However, authors faced two problems: (1) a new group of animals should be included in the experiment for that purpose, (2) The variety of the gathered data so far is enough to suggest a potential better performance of the n-HA/bTCP.
- Na explanation is needed about regarding the particle sizes of autogenous bone and n-HA/bTCP implanted in bone defect area.
Answer:
- Please explain in detail whether n-HA/bTCP is more efficient for bone regeneration than autogenous bone graft.
Answer:
the current literature demonstrates this, where nano-hydroxyapatite promotes better bone regeneration than autogenous bone, mainly because autogenous bone promotes great resorption when compared to nano-biomaterial.
Author needs some wordind and gramar correction.
Answer:
we will review the English with the mdpi team
Round 2
Reviewer 2 Report
Comments and Suggestions for Authors
This study compared bone regeneration by nano-hydroxyapatite/β-tricalcium phosphate (n-HA/β-TCP) composite with autogenous bone grafts.
1. Please describe the form of autogenous bone graft used in this paper. Please check the size at that time
Comments on the Quality of English LanguageThe author needs some wording and grammar corrections
Author Response
Dear
The autogenous bone is removed from the calvaria and kneaded using a chisel until it resembles the granule size of the nano-hydroxyapatite sample, which is around 0.5-1mm.
The answer has been inserted in the methods section in red.
I will carry out the grammar correction with the mdpi team
Sincerely.